# Recent Progress in Circularly Polarized Luminescent Materials Based on Cyclodextrins

**DOI:** 10.3390/polym16152140

**Published:** 2024-07-27

**Authors:** Chengkai Zhou, Weixing Chang, Lingyan Liu, Jing Li

**Affiliations:** 1The State Key Laboratory and Institute of Elemento-Organic Chemistry, College of Chemistry, National Engineering Research Center of Pesticide, Nankai University, Tianjin 300071, China; 1120220449@mail.nankai.edu.cn (C.Z.); changwx@nankai.edu.cn (W.C.); 2National Engineering Research Center of Pesticide, Nankai University, Tianjin 300071, China

**Keywords:** CPL materials, cyclodextrins, host–guest interaction

## Abstract

Circularly polarized luminescence (CPL) materials have been widely used in the fields of bioimaging, optoelectronic devices, and optical communications. The supramolecular interaction, involving harnessing non-covalent interactions between host and guest molecules to control their arrangements and assemblies, represents an advanced approach for facilitating the development of CPL materials and finely constructing and tuning the desired CPL properties. Cyclodextrins (**CDs**) are cyclic natural polysaccharides, which have also been ubiquitous in various fields such as molecular recognition, drug encapsulation, and catalyst separation. By adjusting the interactions between **CDs** and guest molecules precisely, composite materials with CPL properties can be facilely generated. This review aims to outline the design strategies and performance of **CD**-based CPL materials comprehensively and provides a detailed illustration of the interactions between host and guest molecules.

## 1. Introduction

Circularly polarized luminescence (CPL) materials provide crucial support for the development of some fields such as 3D displays, biological imaging, and information storage and encryption [1,2,3,4,5,6]. CPL can be generated through a polarizer and quarter-wave plate; however, the significant energy loss during this process and the bulky setup constrain its applications [7]. Fortunately, scientists have discovered that some substances, including coordinative compounds, organic small molecules, polymers, liquid crystal materials, and so forth, can exhibit CPL activity [8,9,10,11,12,13]. The ability of a substance to emit circularly polarized light is evaluated by the dissymmetry factor (*g_lum_*), defined as
*g_lum_* = 2 (I*_L_* − I*_R_*)/(I*_L_* + I*_R_*)
where I*_L_* and I*_R_* are the intensities of the left and right circularly polarized emission, respectively [14]. *g_lum_* ranges from +2 to −2, corresponding to an optimal emission of left or right-handed CPL, respectively. When *g_lum_* = 0, it indicates that there is an emission of non-circularly polarized light. The *g_lum_* values of all CPL materials developed usually fall within the range of 10^−4^ to 10^−3^ at present. Therefore, achieving a larger *g_lum_* has become the fervent and indefatigable pursuit of thousands upon thousands of scientists.

Supramolecular assembly usually involves non-covalent interactions such as hydrogen bonding, van der Waals forces, and π-π stacking between molecules, enabling host and guest molecules to form stable structures with specific shapes and functionalities under appropriate conditions [15]. The supramolecular assembly has been used as an efficient strategy for the construction of CPL materials. For example, cyclodextrins (**CDs**) are natural macrocyclic structures formed by linking six, seven, or eight D-glucose molecules via α-1,4 glycosidic bonds, called **α-**, **β-**, and **γ-CD**, respectively, which feature a hydrophobic inner cavity and hydrophilic outer surface and have the ability to form inclusion complexes with numerous organic and inorganic compounds, thereby bringing about diverse assemblies with specific properties and functions [16]. With the extensive applications in the fields of chemistry and biology, **CDs** hold significant potential for a wide range of industrial applications [17,18,19].

In addition, the combination of fluorescent dyes with **CDs** to produce composite materials with CPL emission has developed rapidly, and **CD** derivatives are utilized in preparing nanomaterials and nanostructures with CPL performance. The multilevel self-assembly strategy is employed to amplify the *g_lum_* of CPL materials. Host–guest complexes initially form basic unit structures, which can further assemble into advanced structures such as nanospheres, nanotubes, and helical architectures. This process is typically regulated by physical, chemical, and environmental factors, thus enabling the production of materials or devices with specific functions and properties. Under the driving force of host–guest chemistry, the excellent encapsulation and molecular recognition properties of **CDs** provide new insights and methods for the design and fabrication of excellent nanomaterials (Figure 1).

Given the significant role of **CDs** in CPL emission, herein, we review the strategies for constructing CPL materials based on **CDs** in the past decade and prospect their future development. It is believed that this summary of strategies for constructing CPL materials by **CDs** will be informative and helpful for CPL material developers.

## 2. CPL Materials Based on Cyclodextrins

### 2.1. Covalent Grafting of Achiral Fluorescent Dyes with Cyclodextrins

This strategy fundamentally involves covalently linking CDs (host molecules) and fluorescent dyes (guest molecules) via ester or amide bonds. Subsequently, CPL emission is induced by intermolecular or intramolecular non-covalent interactions. Liu et al. [20] choose dansyl as the emitting moiety and **β-CD** as the chiral source to synthesize rotaxane **1** with thermo-reversible CPL performance. Rotaxane **1** shows three different conformations in H_2_O, DMSO, and CH_3_OH (Figure 2a). In a low concentration (3 mM) of aqueous solution, rotaxane **1** forms channel-type **1^A^** with blue emission. However, at high concentrations (10 mM), it can self-assemble into Cage-type **1^B^** with green emission (Figure 2b). The cyclic thermal test for **1^A^** and **1^B^** shows that the fluorescence quantum yield (Φ_f_) of **1^A^** has reversible changes within the range of 90%–80%. In contrast, the Φ_f_ of **1^B^** decreases noticeably from 80% to 72% after cycling. While the fluorescence intensity of the assemblies decreases with increasing temperature, the *g_lum_* of **1^A^** increases and exhibits a reversible thermo-regulated CPL property between 25 °C (*g_lum_* = 0.024) and 90 °C (*g_lum_* = 0.040). However, under the same conditions, the *g_lum_* of **1^B^** continuously decreases with increasing cycle numbers with no thermo-reversibility. In addition, the assembly of the unpotted emitter **1^Agg^** exhibits no significant temperature responsiveness (Figure 2c–g). The variable-temperature powder X-ray diffraction (VT-PXRD) patterns reveal that the flexible framework of **β-CD** undergoes an adaptive reshaping of its hydrogen bonding model with increasing temperature, causing a slight deformation in the cyclic skeleton structure. This limits the nanospace of the **β-CD** cavity for the dansyl moieties, thus achieving thermally enhanced CPL emission.

Shigemitsuk et al. [21] synthesize **α-CD** modified with six pyrenyl groups to obtain pyrene–**CDs 2**. Under the influence of **α-CD**, the pyrene units adopt a chiral arrangement, as is evident by the cotton effect observed in the circular dichroism spectrum (Figure 3a–c). Simultaneously, compound **2** shows a strong CPL signal associated with the excited state emission band, with a maximum *g_lum_* = 1.2 × 10^−2^ (Figure 3d). However, compound **3** with one pyrene-modified derivative of **α-CD** exhibits no CPL signal (Figure 3d). Further spectroscopic studies reveal that the formation of pyrene excimers in a crowded environment is crucial for CPL anisotropy. TD-DFT calculation indicates that the formation of the excimer of compound **2** is ascribed to the nearly parallel pyrene units **P1** and **P2** in the excited state, and the distance between the carbon atoms of **P1** and **P2** decreases from 4.8 Å to 3.8 Å.

Yang et al. [22] further demonstrate that pyrene-modified **γ-CD** derivatives **4**–**6** (Figure 4a,b) can aggregate to nanostructures in water, where the pyrene units are well confined within the enclosed chiral cavity of **γ-CD** (Figure 4d). The host–guest complexes exhibit a significant CPL signal, with a maximum *g_lum_* = 5.3 × 10^−2^ (Figure 4c).

### 2.2. Host–Guest Interaction between Cyclodextrins and Achiral Fluorescent Dyes

Duan et al. [23] report dendritic hydrogel **7**, which can self-assemble into a helical nanofiber (Figure 5a). Additionally, **7** is capable of interacting with **CDs** through host–guest recognition, forming supramolecular structures with CPL emission. To some extent, the CPL signal of hydrogel **7** decreases with increasing pH values and the maximum *g_lum_* is 6.87 × 10^−3^ at pH = 3 (Figure 5b). Due to the significant reduction in strong π-π stacking of the pyrene groups caused by the encapsulation between **7** and **β-CD**, the CPL emission of **7@β-CD** shows a blue shift. Furthermore, the formation of a 2:1 complex of **7@γ-CD** leads to a significant enhancement in CPL emission centered at 480 nm. However, no interaction is observed between **7** and **α-CD** (Figure 5c).

Subsequently, Duan et al. [24] discover that **β-CD** and sodium dodecyl sulfate (**SDS**) can assemble into chiral nanosheets, which further self-assembles into helical microtubes through intermolecular hydrogen bonding (Figure 6). Due to the chiral transfer of **β-CD**, the microtubes loaded with dyes exhibit chirality and CPL emission. Notably, the helical microtube doped with **ThT** shows the strongest CPL emission with *g_lum_* = ±0.1.

Drawing inspiration from Duan et al. [24], Yan et al. [25] introduce a novel temperature-triggered reversible CPL system. The temperature-induced disassembly of the aggregate leads to the release of **β-CD** carrying a fraction of the **ThT** dyes (Figure 7a). According to the H-K rule, they effectively modulate the left- and right-handed CPL emission in the self-assembly system of **ThT**-**SDS@2β-CD** by controlling the temperature. In detail, **ThT-SDS@2β-CD** microtubes tend to co-assemble due to strong electrostatic interactions at 25 °C, resulting in right-handed CPL emission centered at 580 nm. With the increasing temperature, hydrophobic interactions cause **ThT** to be concealed in **β-CD**, while its positive charge remains outside. Consequently, negative circular dichroism signals are induced, as followed by the H-K rule (Figure 7b–d). When the temperature reaches 30 °C, **ThT** attached to **SDS@β-CD** shows right-handed CPL emission concentrated at 580 nm. Meanwhile, as the temperature rises up to 35 °C, the left-handed CPL emission undergoes inversion, indicating that the **ThT** is recognized by **β-CD** due to electrostatic interactions with **SDS**. However, CPL emission disappears as the **ThT-SDS@2β-CD** system is disrupted at 40 °C (Figure 7d–f).

Takashima et al. [26] prepare a series of complexes exhibiting CPL emission concentrated at 480 nm, composed of **γ-CD** and hydrophobic pyrene derivatives. Among them, complex **8** shows the highest *g_lum_* (+2.2 × 10^−3^) and 0.6 (Figure 8). They propose that the hydroxyl group of serine forms hydrogen bonds with **γ-CD**, inducing two pyrenyl groups to twist into a stable chiral excimer.

Liu et al. [27] incorporate cyanostilbene-conjugated gelator **9** into **CDs** to result in extensive CPL emission (Figure 9a). The assemblies show a slight increase in *g_lum_* compared to **9** individually (Figure 9c). Particularly, due to the large cavity of **γ-CD**, the gelator **9** can undergo *Z*-*E* isomerization. At this point, the assembly can form into a nanosphere, and CPL emission becomes silenced. Heating the system causes the assembly to transform into a nanotube; in this case, the complex exhibits CPL emission again (Figure 9b).

After that, Liu et al. [28] synthesize novel non-chiral guest molecule **10**, featuring a conjugated connection between pyrene and adamantane. With dual host–guest interaction sites, compound **10** serves as a platform for exploring different **CD**-induced CPL emissions (Figure 10). Due to the differences in the volume of the pyrene and adamantane groups, as well as their affinity for the **CD** cavity, different types of **CDs** can selectively encapsulate the two parts of guest **10**, thereby resulting in different CPL emissions. As anticipated, **α-CD** is too small to accommodate any portion of **10**, thus precluding CPL emission. At low concentrations, **β-CD** captures the adamantane moiety selectively, with pyrene encapsulation occurring as the concentration increases, leading to opposite CPL emission. On the other hand, **γ-CD** exclusively recognizes the pyrene moiety, thereby adhering to the H-K rule, with the CPL emission remaining unaltered. However, its encapsulation behavior of **10** varies at different concentrations.

Subsequently, Liu et al. [29] discover that **CDs** in the aqueous solution can form Langmuir–Schaefer (LS) films with amphiphilic enantiomer **11** at the interface, exhibiting nanofiber characteristics and CPL activity enhancement (Figure 11a). The chiral synergy observed between **L-11** and its **CD** hosts leads to an amplified circular dichroism, whereas the conflicting molecular chirality between **D-11** and its associated **CD** hosts causes a decrease. However, the 2D LS films can stabilize the aggregation state of **11**, thereby enhancing CPL emission (Figure 11c). In α, β, and **γ-CD** aqueous solutions, **D-11** displays a maximum *g_lum_* of −12.4 × 10^−3^, −9 × 10^−3^, and −10.8 × 10^−3^, accompanied by enhancements of 2.48, 1.80, and 2.16 times, respectively. On the other hand, **L-11** manifests *g_lum_* = 18.4 × 10^−3^, 17.1 × 10^−3^, and 19.2 × 10^−3^, indicating corresponding increases of 3.54, 3.29, and 3.69 times (Figure 11b).

Kitamatsu et al. [30] disclose that incorporating **γ-CD** into chiral bipyrenyl oligopeptides containing stereocenters of the same chirality can control the wavelength, intensity, and signal of stimulated supramolecular CPL. After adding **γ-CD**, **LL-12** to **LL-15** exhibit CPL emission in their photo-excited state (Figure 12 and Table 1). Upon the addition of **γ-CD**, the emission positions of CPL signals from **LL-12** to **LL-14** have no obvious change, but there is a larger blue-shift observed for **LL-15**. While CPL emission remains relatively stable for **LL-13/γ-CD**, significant increases are observed for both **LL-12/γ-CD** and **LL-14/γ-CD**. Additionally, **LL-15/γ-CD** undergoes a transition. In general, the *g_lum_* values of **LL-12/γ-CD** to **LL-15/γ-CD** are either close to or slightly higher than those of **LL-12** to **LL-15**, with **LL-14/γ-CD** showing a significant increase. However, there is no change in the inclusion behavior of another chiral oligopeptide with bipyrenyl groups in **γ-CD**. The supramolecular CPL emission regions for **DD-12** to **DD-14** remain unchanged. In contrast, **DD-15/γ-CD** exhibits the CPL signals in two distinct bands at 434 and 497 nm, attributed to differences in the pyrene overlap. The CPL emission sharply decreases for **DD-13/γ-CD** and **DD-15/γ-CD**, while it is increased for **DD-12/γ-CD** and **DD-14/γ-CD,** and **DD-15/γ-CD** undergoes a transition. The *g_lum_* values of **DD-12/γ-CD** and **DD-15/γ-CD** slightly increase, with **DD-14/γ-CD** displaying a significant increase compared to itself, respectively.

Takashima et al. [31] develop a series of host–guest systems (**16/γ-CD** to **19/γ-CD**) with high Q_Y_ in the solid state and CPL emission. From **16/γ-CD** to 1**8/γ-CD**, the 9-position-substituted anthracene segments can interact with the **γ-CD** cavity, inducing a right-handed chiral distortion (Figure 13a). Consequently, left-handed CPL emission is observed. However, due to the presence of a single-molecule inclusion entity in **19/γ-CD**, its CPL emission is weak (Figure 13b).

### 2.3. CPL Emission System Based on γ-CD-MOF

Liu et al. [32] successfully encapsulate non-chiral luminescent emitters into **γ-CD-MOF**, resulting in CPL emission from violet to red (Figure 14). They find that the sizes of the emitters can influence the polarity of CPL emission. When the emitters are smaller than the **γ-CD** cavity, CPL emission becomes unpredictable. However, when the emitter sizes are comparable to **γ-CD**, a stable positive CPL emission can be achieved. Larger emitters can be selectively encapsulated into **γ-CD-MOF**.

Subsequently, Stoddart et al. [33] further discover that the absence of non-covalent interactions between the fluorophores and **γ-CD** has only a slight weakening effect on the ability of **γ-CD-MOF** to encapsulate them. However, the co-crystallization of **γ-CD** with fluorescent dyes lacking non-covalent interactions may result in **γ-CD** wrapping the dyes in a disordered manner, predominantly forming a cubic **γ-CD-MOF** structure. This process leads to a loss of control over chiral transfer, culminating in random and unpredictable CPL emissions. Moreover, it is disclosed that 1-pyrenecarboxylate anions (**20**) can exhibit significant non-covalent interactions with **γ-CD** in solution and form a helical chiral structure with **γ-CD-HF** (Figure 15a,b). These alignments result in specific optical and thermal properties, such as controllable CPL emission, with *g_lum_* = +3.5 × 10^−3^ for **20/γ-CD-HF**.

### 2.4. Cyclodextrin-Based Rotaxane Systems

Inouye et al. [34] first develop double alkynylpyrene-threaded [4] rotaxane **21**. In aqueous solution, **21** exhibits bright yellow-green excimer emission with Φ_f_ up to 0.37. By comparing the chiral information exhibited in the circular dichroism and CPL spectra of the system, it is found that the chirality in both spectra originates from the same source. Due to the spatial constraints within the **γ-CD** cavity, the alkynylpyrene pair in **21** twists in an asymmetric manner and the chirality is retained in the excited state. Thus, the system displays intense CPL emission with *g_lum_* = −1.5 × 10^−2^ at 480 nm (Figure 16).

Liu et al. [35] employ a mechanical interlocked strategy, utilizing **β-CD** as chiral wheels and a covalent organic framework consisting of **C3** and **Pyr** as axles to construct the 2D polymer and realize a chiral polyrotaxane (**2D CPR**) monolayer with CPL emission (Figure 17). The structures and properties of **2D CPR** can be regulated by the Feed_(W/A)_ (the molar feeding ratios of **β-CD** to **Pyr** during the synthesis). When the Feed_(W/A)_ reaches 8, the CPL emission of **2D CPR** is strongest, resulting in the formation of large-sized high-quality monolayer films of **2D CPR**.

## 3. Conclusions

From the above design strategies, it can be seen that the **CD**-based CPL materials have many advantages. Their structures can be finely tuned through chemical modification, thus influencing their CPL performance and offering increased flexibility in developing specific functional materials compared to other substances. In addition, the cyclic structure of **CDs** facilitates the efficient encapsulation of chiral molecules while effectively transmitting and amplifying chiral signals, which enhances the efficiency of CPL and holds substantial practical significance. Moreover, **CDs** and their derivatives generally exhibit excellent biocompatibility, particularly advantageous in biomedical applications such as bioimaging. **CD**-based materials can also show multiple luminescent properties by combining different fluorescent dyes, as demonstrated by the significant versatility for developing innovative optoelectronic devices and sensors. These materials typically feature robust chemical stability and environmental friendliness while delivering outstanding performance that meets sustainability requirements.

Of course, **CDs** exhibit selectivity and limitations in the size and shape of molecules, which may exclude larger or unconventional shapes. In addition, their performance requires specific conditions to exhibit excellent CPL, and this characteristic is significantly influenced by environmental factors such as solvent and pH, which limits their application in certain scenarios. Despite continuous advancements in research, **CD**-based CPL materials encounter challenges in commercialization and large-scale applications due to issues such as cost, stability, and market acceptance.

In summary, we review the development of **CD**-based CPL materials over the past decade. CPL emission can be generated by modifying **CDs** with non-chiral dyes directly, host–guest interactions, and the assembly of **CDs** with non-chiral dyes. Additionally, the unique cubic chirality of **γ-CD-MOF** cavities can be utilized to encapsulate non-chiral fluorescent dyes. Furthermore, the rotaxane system is also an effective strategy for constructing CPL systems containing **CDs**. Leveraging the inherent capabilities of **CDs** in guest encapsulation and hierarchical self-assembly, the meticulous control over the CPL performance of materials is achievable through nuanced manipulations of temperature, solvent, and pH, as well as other external environmental factors. However, the majority of materials currently exhibit *g_lum_* values in the range of 10^−4^ to 10^−3^, indicating a considerable distance from practical applications. Fortunately, an increasing body of evidence suggests that gel or film systems constructed based on **CDs** often display stronger CPL emission. With the continuous development and refinement of innovative characterization techniques, coupled with the continuously profound exploration of host–guest interactions and supramolecular assembly by researchers, it is anticipated that **CD**-based CPL materials will unveil unparalleled prospects in the domains of optoelectronics and optical materials in the foreseeable future.

## Figures and Tables

**Figure 1 polymers-16-02140-f001:**
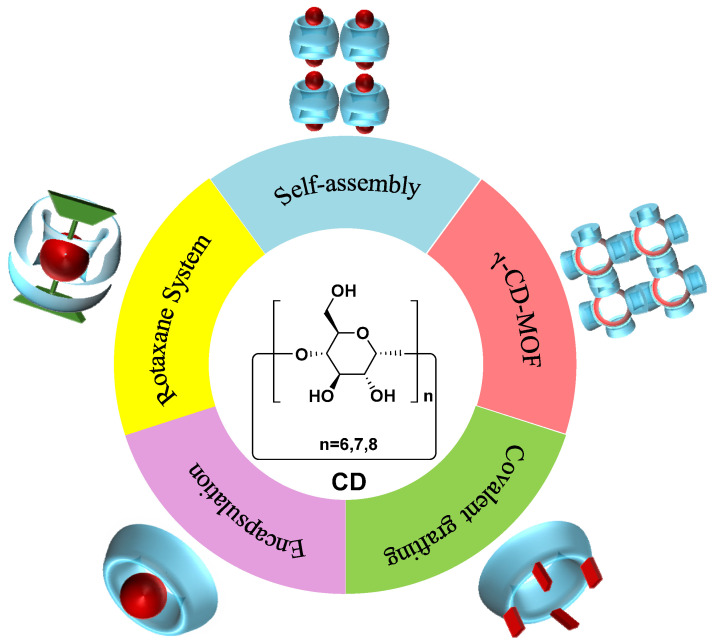
Schematic illustration of design strategies for **CD**-based CPL materials.

**Figure 2 polymers-16-02140-f002:**
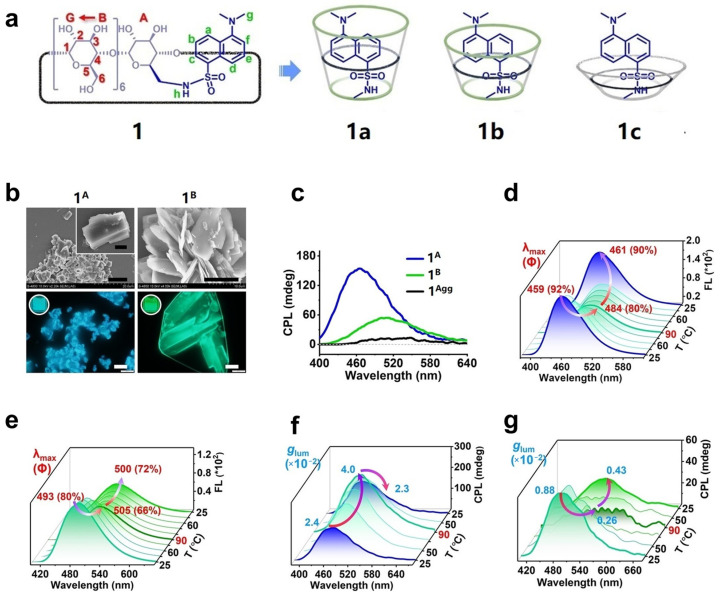
(**a**) Structures of **1a**–**1c**; (**b**) SEM and fluorescence images of **1^A^** and **1^B^**; (**c**) CPL spectra of **1^A^**, **1^B^**, and **1^Agg^**; (**d**,**e**) temperature-dependent FL spectra of **1^A^** and **1^B^**; (**f**,**g**) temperature-dependent *g_lum_* and CPL emission spectra of **1^A^** and **1^B^** [20].

**Figure 3 polymers-16-02140-f003:**
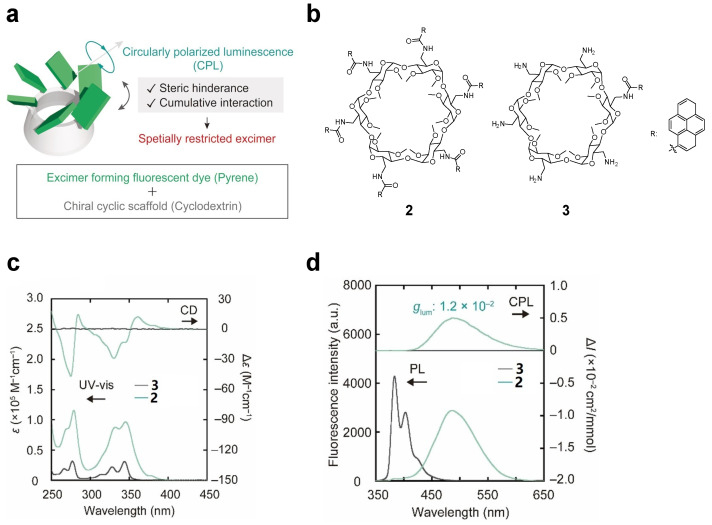
(**a**) CPL emission from pyrene-modified **α-CD** scaffold; (**b**) chemical structures of **2** and **3**; (**c**) circular dichroism (top) and UV-vis absorption spectra (bottom) of **2** and **3**; (**d**) CPL (top) and FL (bottom) spectra of **2** and **3** [21].

**Figure 4 polymers-16-02140-f004:**
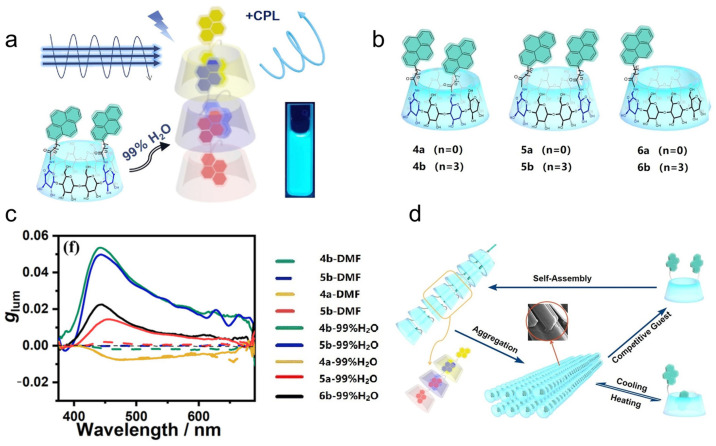
(**a**) The self-assembly of dipyrene-substituted **γ-CD** in H_2_O shows significant CPL activity; (**b**) the structures of pyrene-substituted **γ-CD 4**–**6**; (**c**) the variation in *g_lum_* of **4**–**6** in DMF and 99% H_2_O aqueous solution; (**d**) schematic representation of the self-assembly of **4**–**6** [22].

**Figure 5 polymers-16-02140-f005:**
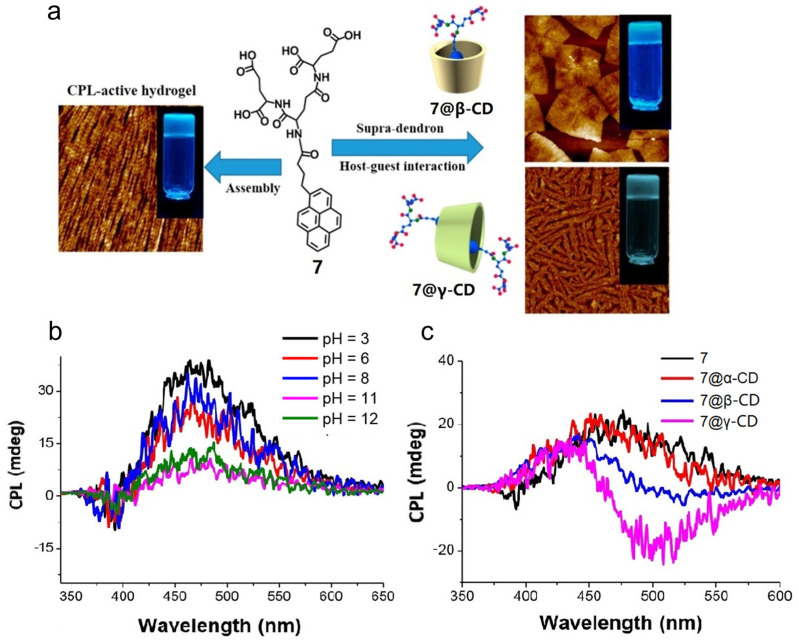
(**a**) AMF images of **7** assembled with **β-CD**, **γ-CD**, and itself; (**b**) CPL spectra of **7** under different pH; (**c**) CPL spectra of **7**, **7@β-CD**, and **7@γ-CD** [23].

**Figure 6 polymers-16-02140-f006:**
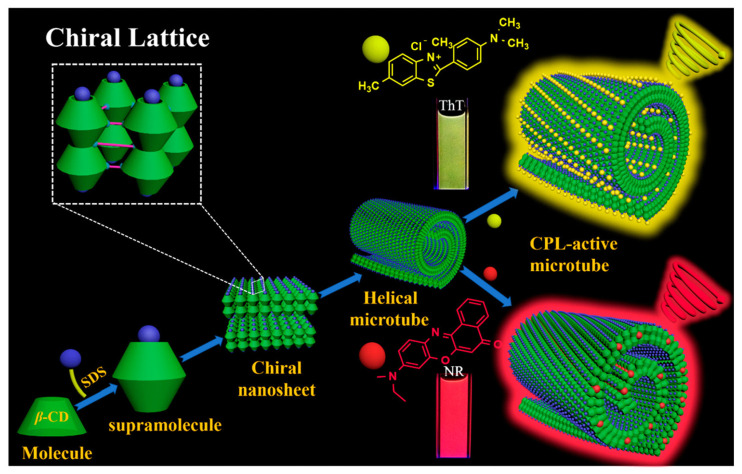
Supramolecular assembly of **β-CD** and **SDS** and CPL emission.

**Figure 7 polymers-16-02140-f007:**
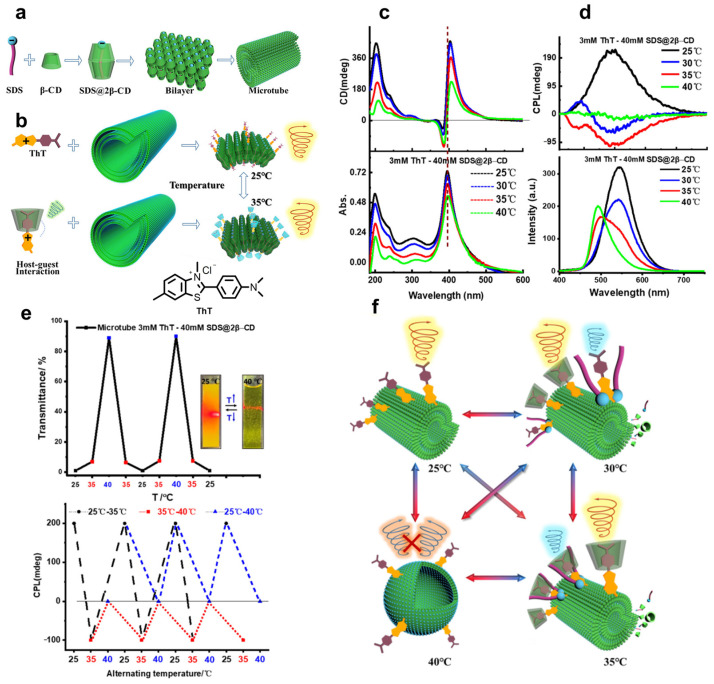
(**a**) Self-assembly of **SDS@2β-CD**; (**b**) reversal of CPL emission of non-chiral dye **ThT** with **SDS@2β-CD** from 25 °C to 35 °C; (**c**) circular dichroism (top) and UV-vis absorption (bottom) spectra of **ThT-SDS@2β-CD** at different temperatures; (**d**) CPL (top) and FL (bottom) spectra of **ThT-SDS@2β-CD** at different temperatures; (**e**) top—variation in the transmittance of the **ThT-SDS@2β-CD** at 650 nm with varying temperature; bottom—switchable CPL inversion cycles (at 590 nm) of 3 mM **ThT**/40 mM **SDS@2β-CD** suspension by alternately changing temperature; (**f**) schematic representation of CPL switching modes of **ThT-SDS@2β-CD** at different temperatures [25].

**Figure 8 polymers-16-02140-f008:**
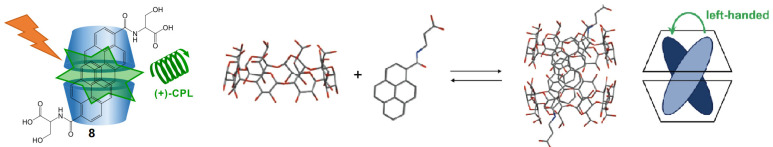
The structure of **γ-CD**-1-(aminocarbonyl) pyrene complex **8** and the schematic illustration for a possible configuration in pyrene/**γ-CD** systems [26].

**Figure 9 polymers-16-02140-f009:**
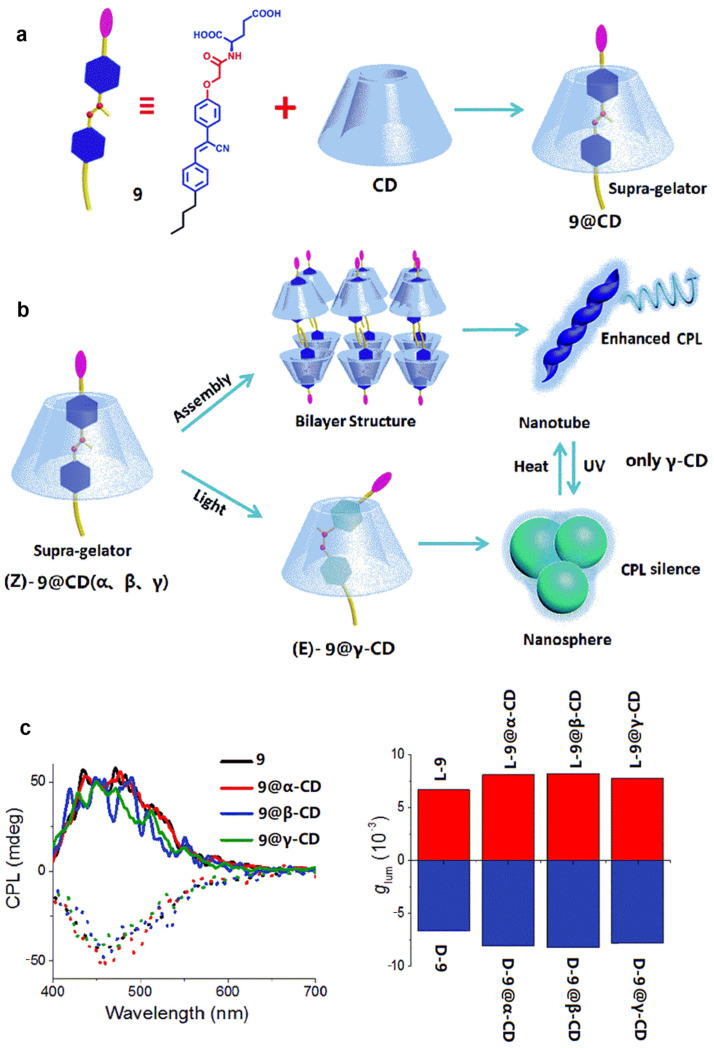
(**a**) Cyanostilbene-conjugated gelator **9** and supra-gelator **9@CD**; (**b**) top—self-assembly of supra-gelator **9@CD**; bottom—morphological transformation from nanotubes to nanospheres of **9@γ-CD**; (**c**) left—CPL spectra of gelator **9** and supra-gelator **9@CD**; right—*g_lum_* of gelator **9** and supra-gelator **9@CD** [27].

**Figure 10 polymers-16-02140-f010:**
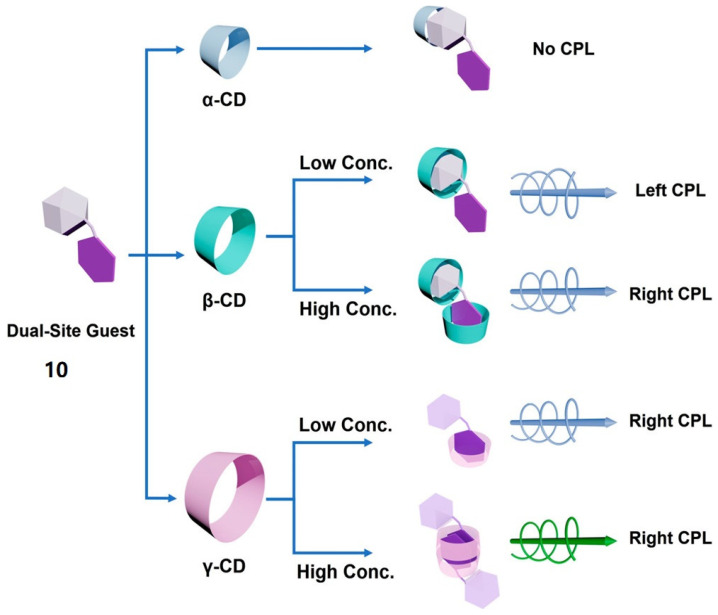
Illustration of selective CPL induction of **10** by different **CDs** [28].

**Figure 11 polymers-16-02140-f011:**
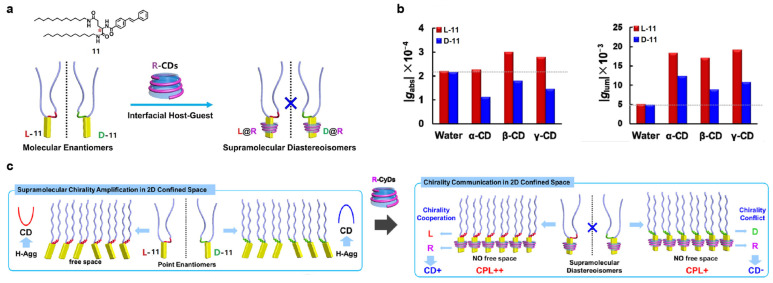
(**a**) Structure of amphiphilic compound **11** and its host–guest interaction with **CDs**; (**b**) changes in *g_abs_* (left) and *g_lum_* (right) after self-assembly of **D/L-11** and **CDs**; (**c**) chirality amplification effect of **11@CD** assemblies [29].

**Figure 12 polymers-16-02140-f012:**
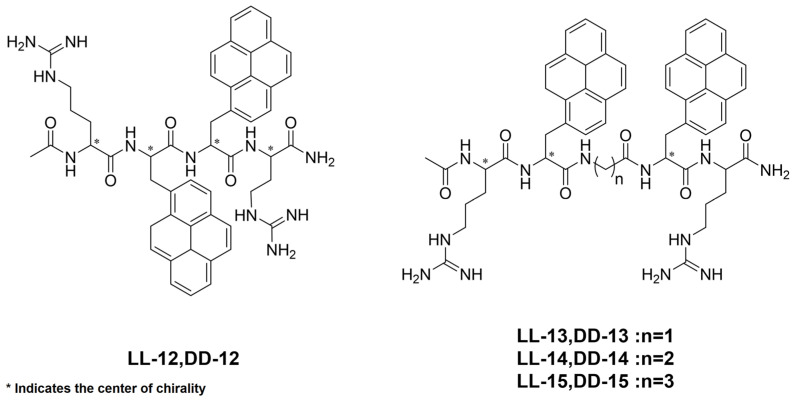
The chemical structures of **DD/LL 12** to **DD/LL 15** [30]. * Indicates the center of chirality.

**Figure 13 polymers-16-02140-f013:**
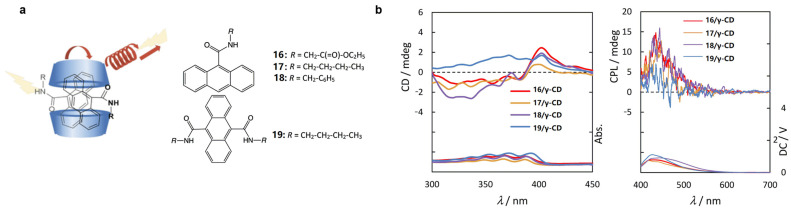
(**a**) Encapsulation of compounds **16**–**19** by **γ-CD**; (**b**) circular dichroism (**left**) and CPL (**right**) spectra of **16**–**19** before and after encapsulation with **γ-CD** [31].

**Figure 14 polymers-16-02140-f014:**
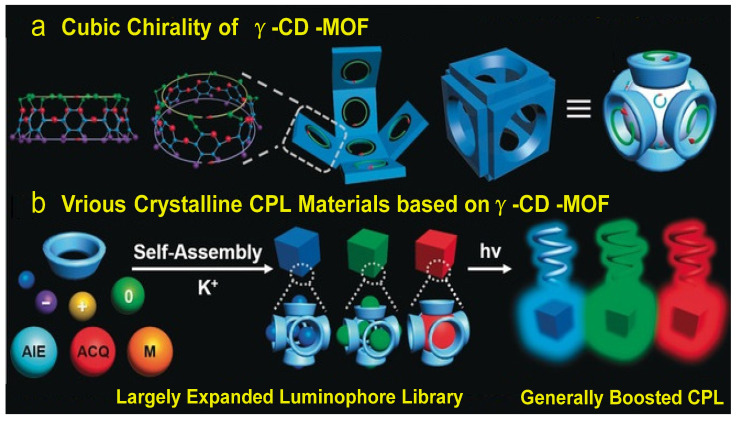
(**a**) Illustration of the cubic chirality of **γ-CD-MOF**; (**b**) various crystalline CPL materials based on **γ-CD-MOF** and non-chiral emitters with enhanced CPL [32].

**Figure 15 polymers-16-02140-f015:**
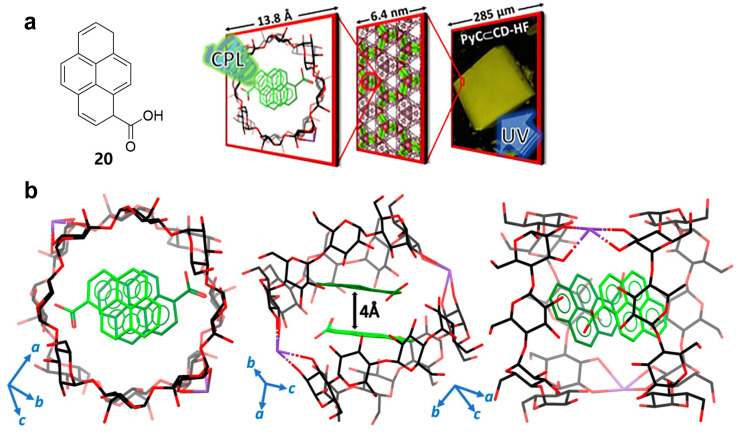
(**a**) Chemical structure of 1-pyrene carboxylic acid **20** and schematic representation of CPL emission for **20/γ-CD-HF**; (**b**) solid-state X-ray diffraction of **20/γ-CD-HF** [33].

**Figure 16 polymers-16-02140-f016:**
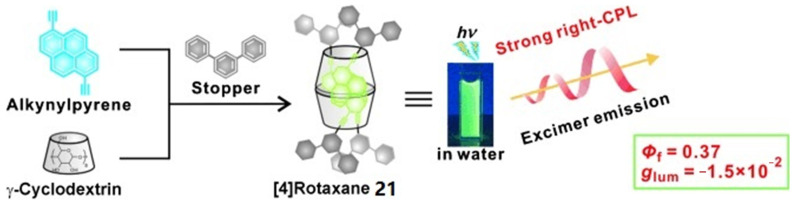
Synthetic scheme of bis(alkynyl) rotaxane **21** and its CPL emission [34].

**Figure 17 polymers-16-02140-f017:**
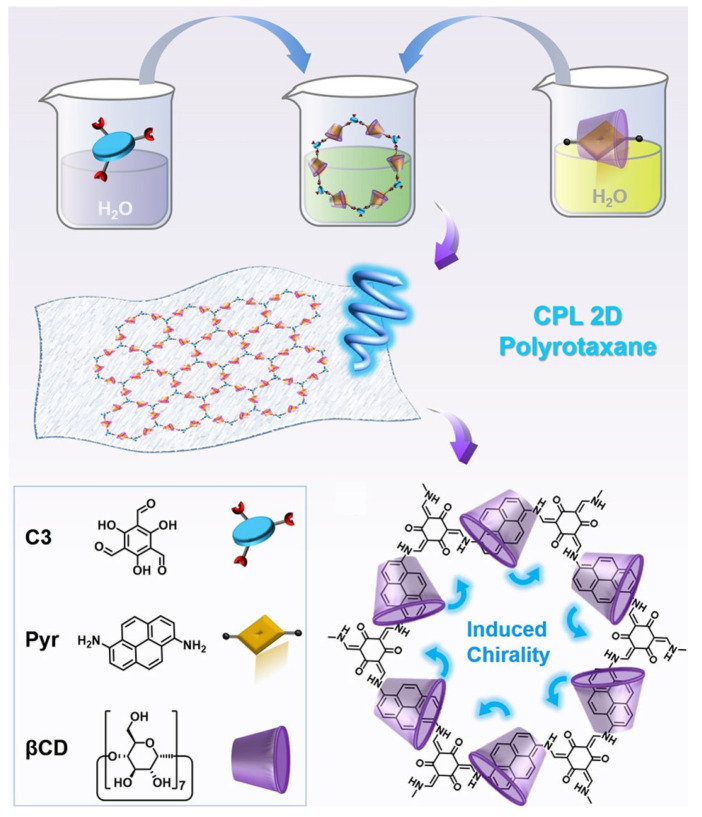
Synthesis process for two-dimensional chiral polyrotaxane (**2D CPR**) with host–guest-induced skeleton chirality, which enables the induction of CPL for **2D CPR** monolayers [35].

**Table 1 polymers-16-02140-t001:** CPL emission and *g_lum_* before and after the addition of **γ-CD** to compound **12**–**15** [30].

Compounds	λ_CPL_ (nm)	*g_lum_* (10^−3^)	Compounds	λ_CPL_ (nm)	*g_lum_* (10^−3^)
**LL-12**	495	+1.8	**LL-12/γ-CD**	468	+3.0
**LL-13**	486	−5.3	**LL-13/γ-CD**	478	−4.7
**LL-14**	496	+1.5	**LL-14/γ-CD**	485	+8.5
**LL-15**	476	−3.3	**LL-15/γ-CD**	444	+6.0
**DD-12**	470	−1.7	**DD-12/γ-CD**	470	−3.8
**DD-13**	483	+5.2	**DD-13/γ-CD**	487	+3.0
**DD-14**	502	−1.6	**DD-14/γ-CD**	483	−14.0
**DD-15**	487	+3.3	**DD-15/γ-CD**	434 497	−1.2 −0.7

## Data Availability

Not applicable.

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
