# Peer review of "Recent Progress in Circularly Polarized Luminescent Materials Based on Cyclodextrins"

_polymers, 2024, doi:10.3390/polym16152140_

Round 1
Reviewer 1 Report
Comments and Suggestions for Authors
The review by Jing Li et al. summarizes recent results in the field of cyclodextrins with CPL applications. Current achievements are well described and analyzed. I think this review appears as a significant contribution into the relevant research and is of sufficient novelty and impact to be published in Polymers. I can only state some minor formal concerns on this work:
1. The text contains plenty of typos and grammar mistakes. Please revise the text carefully.
2. Figure numbering has been failed throughout all the text.
Comments on the Quality of English LanguageThe text contains plenty of typos and grammar mistakes. Please revise the text carefully.
Author Response
For research article
|
Response to Reviewer 1 Comments |
||
|
1. Summary |
|
|
|
Thank you for your valuable feedback on my manuscript. We have carefully addressed the issues you highlighted. We have standardized the tense used in the review to the simple present tense, instead of the simple past tense. All the changes in the review are highlighted with a yellow background. |
||
|
2. Point-by-point response to Comments and Suggestions for Authors |
||
|
Comments 1: The text contains plenty of typos and grammar mistakes. Please revise the text carefully. |
||
|
Response 1: We have carefully checked the grammar and spelling errors in the review and have made the necessary corrections. The following is the revised content. 1.Original content: Recent process in circularly polarized luminescent materials based on cyclodextrins Correction: Recent Progress in Circularly Polarized Luminescent Materials Based on Cyclodextrins Location: Page 1, the line 2 in the manuscript. 2.Original content: Circularly polarized luminescence (CPL) materials have been developed to be widely used in some fields such as bioimaging, optoelectronic devices, and optical communications. Correction: Circularly polarized luminescence (CPL) materials have been widely used in the fields of bioimaging, optoelectronic devices, and optical communications. Location: Page 1, the line 9 in the manuscript. 3.Original content: The supramolecular action, involving harnessing non-covalent interactions between host and guest molecules to control their arrangement and assembly Correction: The supramolecular interaction, involving harnessing non-covalent interactions between host and guest molecules to control their arrangements and assemblies Location: Page 1, the line 12 in the manuscript. 4.Original content: By judiciously adjusting the interactions between CDs and gust molecules, composite materials with CPL activity can be facilely generated. Correction: By adjusting the interactions between CDs and guest molecules precisely, composite materials with CPL property can be facilely generated. Location: Page 1, the line 16 in the manuscript. 5.Original content: This review aims to outline the design strategies and performance of CDs-based CPL materials comprehensively, and provides a detailed examination of the interactions between host and guest molecules. Correction: This review aims to outline the design strategies and performance of CDs-based CPL materials comprehensively and provides a detailed illustration of the interactions between host and guest molecules. Location: Page 1, the line 18 in the manuscript. 6.Original content: In addition, it has been rapidly developed that combining fluorescent dyes with CDs to produce composite materials with CPL property, and CD derivatives are utilized in preparing nanomaterials and nanostructures with CPL activity. Correction: This review aims to outline the design strategies and performance of CDs-based CPL materials comprehensively and provides a detailed illustration of the interactions between host and guest molecules. Location: Page 1, the line 18 in the manuscript. 7.Original content: Circularly polarized luminescent (CPL) materials provide crucial supports for the development of fields such as 3D display, biological imaging, and information storage and encryption. Correction: Circularly polarized luminescent (CPL) materials provide crucial supports for the development of some fields such as 3D display, biological imaging, and information storage and encryption. Location: Page 2, the line 24 in the manuscript. 8.Original content: where IL and IR are the intensities of the left and right circularly polarized emissions, respectively. Correction: In this formula, IL and IR are the intensities of the left and right circularly polarized emission, respectively. Location: Page 2, the line 32 in the manuscript. 9.Original content: When glum = 0, it indicates emission of non-circularly polarized light. Correction: When glum = 0, it indicates that there is emission of non-circularly polarized light. The glum values of all CPL materials developed are usually fall within the range of 10-4 to 10-3 at present. Location: Page 1, the line 35 in the manuscript. 10.Original content: This process serves as an effective strategy for constructing CPL materials. Correction: The supramolecular assembly has been used as an efficient strategy for construction of CPL materials. Location: Page 1, the line 41 in the manuscript. 11.Original content: which feature a hydrophobic inner cavity and a hydrophilic outer surface, and have the ability to form inclusion complexes with numerous organic and inorganic compounds, thereby bringing about diverse assemblies with specific properties and behaviors. Correction: which feature hydrophobic inner cavity and hydrophilic outer surface and have the ability to form inclusion complexes with numerous organic and inorganic compounds, thereby bringing about diverse assemblies with specific properties and functions. Location: Page 1, from the line 44 to line 47 in the manuscript. 12.Original content: With the extensive applications in the fields of chemistry, biology, and materials science, CDs hold significant potential for a wide range of industrial applications. Correction: With the extensive applications in the fields of chemistry and biology, CDs hold significant potential for a wide range of industrial applications. Location: Page 1, from the line 44 to line 47 in the manuscript. 13.Original content: it has been rapidly developed that combining fluorescent dyes with CDs to produce composite materials with CPL property, and CD derivatives are utilized in preparing nanomaterials and nanostructures with CPL activity. Correction: it has been rapidly developed that combining fluorescent dyes with CDs to produce composite materials with CPL emission, and CDs derivatives are uti-lized in preparing nanomaterials and nanostructures with CPL performance. Location: Page 1, from the line 50 to line 52 in the manuscript. 14.Original content: Given the significant role of CDs in CPL emissions, we herein reviewed the strat-egies for constructing CPL materials based on CDs in the past decade and prospected their future development. Correction: Given the significant role of CDs in CPL emission, herein, we reviewed the strat-egies for constructing CPL materials based on CDs in the past decade and prospected their future development. Location: Page 2, from the line 63 in the manuscript. 15.Original content: Liu et al. chose dansyl as the emitting moiety and β-CD as the chiral source to synthesize the CPL active rotaxane 1 with thermo-reversible properties. Correction: Liu et al. choose dansyl as the emitting moiety and β-CD as chiral source to synthesize the rotaxane 1 with thermo-reversible CPL performance. Location: Page 2, from the line 73 in the manuscript. 16.Original content: At low concentration (3 mM) and high concentration (10 mM) in aqueous solution, 1 self-assembled into Channel-type 1A with blue emission and Cage-type 1B with green emission, respectively. Correction: In a low concentration (3 mM) of aqueous solution, the rotaxane 1 forms Channel-type 1A with blue emission. However, at high concentration (10 mM), it can self-assemble into Cage-type 1B with green emission. Location: Page 3, from the line 73 to line 77 in the manuscript. 17.Original content: While the fluorescence intensity of the assemblies decreased with increasing temperature, the glum of 1A increases, exhibiting reversible thermo-regulated CPL properties between 25 °C (glum = 0.024) and 90 °C (glum = 0.040). Correction: While the fluorescence intensity of the assemblies decreases with increasing temperature, the glum of 1A increases and exhibits a reversible thermo-regulated CPL property between 25 °C (glum = 0.024) and 90 °C (glum = 0.040). Location: Page 3, from the line 80 to line 82 in the manuscript. 18.Original content: However, under the same conditions, the glum of 1B continuously decreased with increasing cycle numbers. Additionally, the assembly of the unpotted emitter the 1Agg exhibited no significant temperature responsiveness. Correction: However, under the same conditions, the glum of 1B continuously decreases with increasing cycle numbers with no thermo-reversibility. In addition, the assembly of the unpotted emitter 1Agg exhibits no significant temperature responsiveness. Location: Page 3, the line 84 in the manuscript. 19.Original content: This limited the nano space of the CD framework for the dansyl moiety, thus achieving thermally enhanced CPL properties. Correction: This limits the nano space of the β-CD cavity for the dansyl moieties, thus achieving thermally enhanced CPL emission. Location: Page 3, the line 88 in the manuscript.\ 20.Original content: (d-e) CPL changes of 1A and 1B between 25-90 °C; (f-g) glum changes of 1A and 1B between 25-90 °C Correction: (d-e) Temperature-dependent FL spectra of 1A and 1B; (f-g) Temperature-dependent glum and CPL emission spectra of 1A and 1B Location: Page 3, the line 92 and line 93 in the manuscript. 21.Original content: Shigemitsuk et al. synthesized α-CD modified with six pyrene units 2. Correction: Shigemitsuk et al. synthesize α-CD modified with six pyrenyl groups to obtain pyrene–CDs 2. Location: Page 3, the line 94 and line 95 in the manuscript. 22.Original content: Under the influence of α-CD, the pyrene units adopted a chiral arrangement, as evidenced by the cotton effect observed in the circular dichroism spectra. Correction: Under the influence of α-CD, the pyrene units adopt a chiral arrangement, as is evident by the cotton effect observed in the circular dichroism spectrum. Location: Page 3, the line 95 and line 96 in the manuscript. 23.Original content: However, the compound 3, only one pyrene-modified derivative of α-CD, exhibited no CPL signal. Correction: However, compound 3 with one pyrene-modified derivative of α-CD exhibits no CPL signal. Location: Page 3, the line 99 in the manuscript. 24.Original content: TD-DFT calculations indicated that the formation of the dimer of compound 2 was due to the pyrene units P1 and P2 becoming nearly parallel in the excited state, and the distance between the carbon atoms of P1 and P2 was decreased from 4.8 Å to 3.8 Å. Correction: TD-DFT calculation indicates that the formation of the excimer of compound 2 is ascribed to the nearly parallel of pyrene units P1 and P2 in the excited state, and the distance between the carbon atoms of P1 and P2 decreases from 4.8 Å to 3.8 Å. Location: Page 4, the line 101 and line 104 in the manuscript. 25.Original content: Yang et al. further demonstrated that pyrene-modified γ-CD derivatives 4-6 could aggregate to form nanostructures in water. Correction: Yang et al. further demonstrate that pyrene-modified γ-CD derivatives 4-6 could aggregate to nanostructures in water. Location: Page 5, the line 114 in the manuscript. 26.Original content: Notably, the HM system doped with ThT showed the strongest CPL emission with glum = ±0.1. Correction: Notably, the helical microtube doped with ThT shows the strongest CPL emission with glum = ±0.1. Location: Page 5, the line 133 in the manuscript. 27.Original content: they effectively modulated the left- and right-handed of CPL emission in the system of self-assembly of ThT and SDS@2β-CD by controlling the temperature. Correction: they effectively modulate the left- and right-handed CPL emission in the self-assembly system of ThT-SDS@2β-CD by controlling the temperature. Location: Page 6, the line 141 in the manuscript. 28.Original content: Consequently, negative circular dichroism signals were produced, as followed the H-K rule. Correction: Consequently, negative circular dichroism signals were produced, as followed by H-K rule. Location: Page 6, the line 146 in the manuscript. 29.Original content: They proposed that the hydroxyl group of serine formed the hydrogen bonds with γ-CD, thereby fixing the twisted structure and stabilizing the chiral dimeric structure. Correction: They propose that the hydroxyl group of serine forms hydrogen bonds with γ-CD, inducing two pyrenyl groups to twist into a stable chiral excimer. Location: Page 6, the line 137 and line 138 in the manuscript. 30.Original content: Liu et al. discovered that incorporating cyanostilbene conjugated gelator 9 into CDs resulted in extensive CPL emission. Correction: Liu et al. incorporate cyanostilbene conjugated gelator 9 into CDs to result in extensive CPL emission. Location: Page 8, the line 168 in the manuscript. 31.Original content: L-11 manifested glum = 18.4×10-3, 17.1×10-3, and 19.2×10-3, indicating the correspondingly increase of 3.54, 3.29, and 3.69 times Correction: L-11 manifests glum = 18.4×10-3, 17.1×10-3, and 19.2×10-3, indicating the corresponding increase of 3.54, 3.29, and 3.69 times. Location: Page 10, the line 203 in the manuscript. 32.Original content: Kitamatsu et al. discovered that incorporating γ-CD into chiral bipyrenyl oligopeptides containing stereocenters of the same chirality enabled the control of the wavelength, intensity, and sign of stimulated supramolecular CPL. Correction: Kitamatsu et al. disclose that incorporating γ-CD into chiral bipyrenyl oligopeptides containing stereocenters of the same chirality can control the wavelength, intensity and signal of stimulated supramolecular CPL. Location: Page 10, the line 209 in the manuscript. 33.Original content: Stoddart et al. further discovered that the absence of non-covalent interactions between the fluorophores and γ-CD had no significant weaken on the ability of γ-CD-MOF to encapsulate them. Correction: Stoddart et al. further discover that the absence of non-covalent interactions between the fluorophores and γ-CD has only a slight weakening effect on the ability of γ-CD-MOF to encapsulate them. Location: Page 12, the line 250 in the manuscript. 34.Original content: they found that 1-pyrenecarboxylate anion (20) could exhibit significant non-covalent interactions with γ-CD in solution and formed helical chiral structures γ-CD-HF. Correction: it is disclosed that 1-pyrenecarboxylate anion (20) can exhibit significant non-covalent interaction with γ-CD in solution and form a helical chiral structure with γ-CD-HF. Location: Page 12, the line 256 in the manuscript. 35.Original content: Inouye et al. developed the first biphenyl-based bisalkynyl rotaxane 21. Correction: Inouye et al. firstly develop a double alkynylpyrene-threaded [4]rotaxane 21. Location: Page 13, the line 265 in the manuscript. 36.Original content: Liu et al.employed a mechanical interlocked strategy, utilizing β-CD as chiral wheels and a covalent organic framework consisting of C3 and Pyr as axle for the 2D polymer, to realize a chiral polyrotaxane (2D CPR) monolayer with CPL emission. Correction: Liu et al. employ a mechanical interlocked strategy, utilizing β-CD as chiral wheels and a covalent organic framework consisting of C3 and Pyr as axles to construct the 2D polymer and realize a chiral polyrotaxane (2D CPR) monolayer with CPL emission Location: Page 14, the line 276 in the manuscript. 37.Original content: When the Feed(W/A) reached to 8, the CPL emission of 2D CPR was at its strongest, resulting in the formation of large-sized high-quality monolayer films of 2D CPR. Correction: When the Feed(W/A) reaches to 8, the CPL emission of 2D CPR is strongest, resulting in the formation of large-sized high-quality monolayer films of 2D CPR. Location: Page 14, the line 280 in the manuscript. |
||
|
Comments 2: Figure numbering has been failed throughout all the text. |
||
|
Response 2: Thank you for pointing out this error. We have noticed the issue with the image numbering, which occurred due to the interruption of hyperlinks between the images and their numbers during the manuscript revision. We have renumbered the images. Please see the main text of the manuscript, with the changes highlighted in yellow background.
|
||

Reviewer 2 Report
Comments and Suggestions for Authors
This review is on luminescent materials able to emit polarized light based in ciclodextrins as scaffold. The latter are macrocyclic structures which can host and bond to a diversity of fluorescent dyes to produce tailored circularly polarized light. As a review this manuscript presents a brief Introduction section with the basics of circularly polarized luminescent (CPL) materials and the ciclodextrins scaffold features. Then Section 2 of the manuscript presents a summary of the main results of 15 recently published papers (most of them from the last five years) addressing materials with covalent bonding between ciclodextrins and dyes, host-guest interactions, ciclodextrins based MOFs for the dye encapsulation as well as ciclodextrins-rotaxane systems. CPL is a topic of potential for different applications, so this review could of interest for readers of Polymers journal, mainly those searching for condensated information of optical features rather than chemical synthesis protocols. The summary of results from those 15 published papers is robust and might result useful as mini-monograph. Nevertheless, as it is focused in the specific ciclodextrins scaffold, some readers might find this mini-monograph of limited interest compared to more global CPL reviews.
English, style and organization is also sound. Please correct the typos in numbering figures. This review should include a section with a broader description of advantages and drawbacks compared with other scaffolds used in CPL materials.
Author Response
For review article
|
Response to Reviewer 2 Comments
|
||
|
1. Summary |
|
|
|
Thank you for your valuable feedback on this review. Based on your suggestions, we have emphasized the advantages of the cyclodextrin framework compared to other circularly polarized luminescent materials in the conclusion section. Below are the specific modifications made according to your recommendations. |
||
|
Comments 1: English, style and organization is also sound. Please correct the typos in numbering figures. Response 1: We have noticed the issue with the image numbering, which occurred due to the interruption of hyperlinks between the images and their numbers during the manuscript revision. We renumbered the images again. Please check the main text of the manuscript, with the changes highlighted in yellow background. Comments 2: This review should include a section with a broader description of advantages and drawbacks compared with other scaffolds used in CPL materials. |
||
|
Response 2: According to your suggestions, we have added a discussion in the conclusion section of the review, highlighting the advantages of CDs-based CPL materials compared to other materials, to further emphasize the significance of this review. Correction: From the above designing strategies, it can be seen that the CDs-based CPL materials have many advantages. Their structures can be finely tuned through chemical modification, thus influencing their CPL performance and offering increased flexibility in developing specific functional materials compared to other substances. In addition, the cyclic structure of CDs facilitates efficient encapsulation of chiral molecules while effectively transmitting and amplifying chiral signals, which enhances the efficiency of CPL and holds substantial practical significance. Moreover, CDs and their derivatives generally exhibit excellent biocompatibility, particularly advantageous in biomedical applications such as bioimaging. CDs-based materials can also show multiple luminescent properties by combining different fluorescent dyes, as demonstrates significant versatility for developing innovative optoelectronic devices and sensors. These materials typically feature robust chemical stability and environmental friendliness while delivering outstanding performance that meets sustainability requirements. Of course, CDs exhibit selectivity and limitations in the size and shape of molecules, which may exclude larger or unconventional shapes. Besides, their performance requires specific conditions to exhibit excellent CPL, and this characteristic is significantly influenced by environmental factors such as solvent and pH, which limits their application in certain scenarios. Despite of continuous advancements in research, CDs-based CPL materials encounter challenges in commercialization and large-scale applications due to issues such as cost, stability, and market acceptance. Location: page 14, the first and second paragraph of conclusion.
|
||

Reviewer 3 Report
Comments and Suggestions for Authors
This review article need to be polished more. Please check the spellings and unnecessary spaces in the entire manuscript. Text did not include the figure numbers. Please include figure numbers in the text. It would be easy for the readers.
Please see the comments below.
1. In page 2, under CPL materials based on cyclodextrins, author should mention in the text about covalent bond between host cyclodextrin and guest molecules.
2. What about the quantum yields?
3. What are 0a, 0b, 0c-g? There is nothing mentioned in the Figure 2 caption.
4. In page 4, figure 3, author should shown the compound 3 structure. It would be easy comparison for readers between compound 2 and 3.
5. In page 3, paragraph 2, what are 0a-c, 0d denoted? There are not mentioned in Figure 3 captions.
6. In page 5, paragraph 1. author mentioned the wrong reference number and in figure 5 caption too. Also check 0a, 0b and 0c.
7. In page 7, instead of mentioning they (24) please mention name of the author of the particular reference. Also, what is 0 represent in the text?
8. In page 7, paragraph 2, Drawing inspiration from Duan. What is the reference number for Duan paper. Instead of mentioning Duan please mention Duan et.al.
9. Similarly as I mentioned earlier, please check every paragraph about 0a, 0b, 0c etc. It does not make. any sense in the text and captions.
10. In page 10, paragraph 1, they [20]. Mention the author name.
11. In all the references, corresponding author names were mentioned. But in page 13, paragraph 2, why did you mention Inouye?
12. Also, more detail discussion needed for the reference [34].
Author Response
For review article
|
Response to Reviewer 3 Comments
|
||
|
1. Summary |
|
|
|
Thank you for your suggestion of the review, and we have changed the references, image numbers, spelling, and sentences in the manuscript based on your comments. We have standardized the tense used in the review to the simple present tense, instead of the simple past tense. All the changes in the review are highlighted with a yellow background. |
||
Comments 1: In page 2, under CPL materials based on cyclodextrins, author should mention in the text about covalent bond between host cyclodextrin and guest molecules.
Response 1: We added a section before the first paragraph of 2.1 discussing the basic types of covalent bonds.
Correction: This strategy fundamentally involves covalently linking CDs (host molecules) and fluorescent dyes (guest molecules) via ester or amide bonds. Subsequently, CPL emission is induced by intermolecular or intramolecular non-covalent interactions.
Location: Page 2, from the line 69 to line 71 in the manuscript.
Comments 2: What about the quantum yields?
Response 2: In the discussion reference [20], we have included the changes in fluorescence quantum yield (Φf) of the system during temperature variation.
Correction: The cyclic thermal test for 1A and 1B shows that the fluorescence quantum yield (Φf) of 1A has reversible changes within the range of 90%-80%. In contrast, the Φf of 1B decreases noticeably from 80% to 72% after cycling.
Location: Page 3, from the line 77 to line 79 in the manuscript.
Comments 3: What are 0a, 0b, 0c-g? There is nothing mentioned in the Figure 2 caption.
Response 3: Thank you for pointing out this error. We have noticed the issue with the image numbering, which occurred due to the interruption of hyperlinks between the images and their numbers during the manuscript revision. We have renumbered the images. Please see the main text of the manuscript, with the changes highlighted in yellow background.
Comments 4: In page 4, figure 3, author should show the compound 3 structure. It would be easy comparison for readers between compound 2 and 3.
Response 4: Thank you for your advice. We have added the structure of compound 3 in Figure 3b.
Correction:
Location: Figure 3b.
Comments 5: In page 3, paragraph 2, what are 0a-c, 0d denoted? There are not mentioned in Figure 3 captions.
Response 5: Thank you for identifying this error. We have renumbered the references.
Comments 6: In page 5, paragraph 1. author mentioned the wrong reference number and in figure 5 caption too. Also check 0a, 0b and 0c.
Response 6: Thank you for identifying this error. We have reformatted the review and renumbered the references.
Comments 7: In page 7, instead of mentioning they (24) please mention name of the author of the particular reference. Also, what is 0 represent in the text?
Response 7: We apologize for the confusion. "They" refers to the work of Duan et al. In reference [23], we mentioned Duan et al., so we used "They" in reference [24] as a substitute. We have now replaced "They" with "Duan et al." During the text correction process, the hyperlinks between the images and their captions were disrupted, causing many image numbers to display as 0. We apologize for any inconvenience this may have caused in your reading. Thank you for pointing out this error. We have renumbered all the images, which can be seen in the red text sections of the manuscript.
Comments 8: In page 7, paragraph 2, Drawing inspiration from Duan. What is the reference number for Duan paper. Instead of mentioning Duan please mention Duan et.al.
Response 8: "Drawing inspiration from Duan" refers to the work in reference [24]. Since it was cited earlier, we did not cite it again. Following your suggestion, we have now added the citation for reference [24] and replaced "Duan" with "Duan et al."
Comments 9: Similarly, as I mentioned earlier, please check every paragraph about 0a, 0b, 0c etc. It does not make. any sense in the text and captions.
Response 9: Thank you for pointing out this error. We have renumbered all the images, which can be seen in the red text sections of the manuscript.
Comments 10: In page 10, paragraph 1, they [20]. Mention the author name,
Response 10: Following your suggestion, we have replaced "They" with "Liu et al."
Comments 11: In all the references, corresponding author names were mentioned. But in page 13, paragraph 9, why did you mention Inouye?
Response 11: Professor Masahiko Inouye is the first and corresponding author of reference [34]. We can see the authors' order and information on the journal website as shown in the picture.
Comments 12: Also, more detail discussion needed for the reference [34].
Response 12: We provided a more detailed interpretation of reference [34], primarily adding an explanation of the origin of the system's chirality.
Original content: Inouye et al. [34] developed the first biphenyl-based bisalkynyl rotaxane 21. Due to the asymmetric twisting of the two stacked biphenyls within the rotaxane, CPL emission was observed with a maximum glum = 1.5 × 10-2 (0).
Correction: Inouye et al. [34] firstly develop a double alkynylpyrene-threaded [4]rotaxane 21. In aqueous solution, 21 exhibits bright yellow-green excimer emission with Φf up to 0.37. By comparing the chiral information exhibited in the circular dichroism and CPL spectra of the system, it is found that the chirality in both spectra originates from the same source. Due to the spatial constraints within the γ-CD cavity, the alkynylpyrene pair in 21 twists in an asymmetrically manner and the chirality is retained in the excited state. Thus, the system displays intense CPL emission with glum = -1.5 × 10-2 at 480 nm (Figure 16).
Location: Page 13, from the line 265 to line 271 in the manuscript.

Round 2
Reviewer 2 Report
Comments and Suggestions for Authors
In my opinion, the revised manuscript can be accepted for publication